# Therapeutic Effects of Curcumin Derivatives against Obesity and Associated Metabolic Complications: A Review of In Vitro and In Vivo Studies

**DOI:** 10.3390/ijms241814366

**Published:** 2023-09-21

**Authors:** Marakiya T. Moetlediwa, Rudzani Ramashia, Carmen Pheiffer, Salam J. J. Titinchi, Sithandiwe E. Mazibuko-Mbeje, Babalwa U. Jack

**Affiliations:** 1Biomedical Research and Innovation Platform, South African Medical Research Council, Cape Town 7505, South Africa; marakiya.moetlediwa@mrc.ac.za (M.T.M.); rudzani.ramashia@mrc.ac.za (R.R.); carmen.pheiffer@mrc.ac.za (C.P.); 2Department of Biochemistry, North-West University, Mmabatho 2745, South Africa; sithandiwe.mazibukombeje@nwu.ac.za; 3Centre for Cardio-Metabolic Research in Africa (CARMA), Division of Medical Physiology, Faculty of Medicine and Health Sciences, University of Stellenbosch, Cape Town 7505, South Africa; 4Department of Obstetrics and Gynaecology, Faculty of Health Sciences, University of Pretoria, Pretoria 0001, South Africa; 5Department of Chemistry, Faculty of Natural Science, University of the Western Cape, Bellville 7535, South Africa; stitinchi@uwc.ac.za

**Keywords:** curcumin, curcumin derivatives, synthetic compounds, curcuminoids, obesity, obesity-associated complications

## Abstract

Obesity is a major cause of morbidity and mortality globally, increasing the risk for chronic diseases. Thus, the need to identify more effective anti-obesity agents has spurred significant interest in the health-promoting properties of natural compounds. Of these, curcumin, the most abundant and bioactive constituent of turmeric, possesses a variety of health benefits including anti-obesity effects. However, despite its anti-obesity potential, curcumin has demonstrated poor bioavailability, which limits its clinical applicability. Synthesizing curcumin derivatives, which are structurally modified analogs of curcumin, has been postulated to improve bioavailability while maintaining therapeutic efficacy. This review summarizes in vitro and in vivo studies that assessed the effects of curcumin derivatives against obesity and its associated metabolic complications. We identified eight synthetic curcumin derivatives that were shown to ameliorate obesity and metabolic dysfunction in diet-induced obese animal models, while five of these derivatives also attenuated obesity and associated metabolic complications in cell culture models. These curcumin derivatives modulated adipogenesis, lipid metabolism, insulin resistance, steatosis, lipotoxicity, inflammation, oxidative stress, endoplasmic reticulum stress, apoptosis, autophagy, fibrosis, and dyslipidemia to a greater extent than curcumin. In conclusion, the findings from this review show that compared to curcumin, synthetic curcumin derivatives present potential candidates for further development as therapeutic agents to modulate obesity and obesity-associated metabolic complications.

## 1. Introduction

Obesity is considered a public health crisis, with its prevalence rapidly increasing over the past few decades [1]. Currently, it is estimated that 13% (650 million) of the global adult population is obese (body mass index [BMI] ≥ 30 kg/m^2^), with the prevalence projected to reach 20% by 2045 if no effective therapeutic interventions are established [2]. Obesity is associated with metabolic disorders including insulin resistance [3], inflammation [4], lipotoxicity [5], dyslipidemia [6], and non-alcoholic fatty liver disease (NAFLD) [7]. As such, obesity is considered the major risk factor for the development of non-communicable diseases such as type 2 diabetes (T2D) [8], cardiovascular disease (CVD) [9], and certain types of cancers [10]. Effective therapeutic agents are needed to decrease the burden of obesity and its associated complications. 

Obesity is characterized by the excessive accumulation of fat due to an imbalance between energy intake and energy expenditure [11]. Consequently, lifestyle modifications that include reduced caloric consumption and increased physical activity are the primary anti-obesity treatment strategies [12]. However, adherence to lifestyle modifications is poor, leading to an increased reliance on pharmaceutical agents to treat obesity. The use of anti-obesity drugs is plagued with adverse effects such as cardiovascular effects, stroke, cancer, and psychiatric disorders among others [13], and many have been withdrawn from the market [14]. Bariatric surgery, although shown to be an effective anti-obesity strategy, is recommended for morbidly obese individuals only (BMI ≥ 40 kg/m^2^) and is associated with post-surgical complications [15]. This indicates the need for the development of safer and more effective therapeutic agents to overcome the limitations associated with current anti-obesity intervention strategies. 

Accumulating evidence has demonstrated that natural compounds derived from plant sources are effective in treating obesity and its associated metabolic complications. One such natural compound is curcumin, the main bioactive and abundant compound found in the rhizome of the turmeric (*Curcuma longa*) plant, which is used for culinary and medicinal purposes globally [16]. Curcumin has been extensively studied for many years with several beneficial health effects being demonstrated. Of relevance to this review, studies have reported antioxidant and anti-inflammatory [17], hepatoprotective [18], cardioprotective [19], anti-diabetic [20], and anti-obesity [21] properties, without any toxicity being displayed [16]. Despite curcumin’s potential as an anti-obesity therapeutic, its poor bioavailability due to its low absorption, rapid metabolism, and extensive elimination following oral administration limits its clinical applicability [22,23]. To address these challenges, research efforts have focused on the development and evaluation of curcumin derivatives as strategies to improve bioavailability and biological efficacy [24]. Curcumin derivatives, which are structural analogs of curcumin, have been shown to display an improved pharmacokinetic profile, enhanced bioavailability, and improved delivery to target sites when compared to the parent curcumin compound [25,26,27]. In addition, these derivatives possess anti-cancer [28], anti-diabetic [29], antioxidant [30], and anti-inflammatory [31] efficacy. 

Several experimental studies have evaluated the effects of curcumin derivatives to treat obesity and its associated metabolic complications. However, there is no clear indication of the efficacy of these compounds especially when compared to curcumin and their potential clinical application. Hence, in this review, we summarize and critically assess in vitro and in vivo studies that have evaluated the therapeutic efficacy of curcumin derivatives against obesity and associated metabolic disorders. Moreover, if data were available, we compared the bioactivity of curcumin and its derivatives in these experimental models. The search strategy was developed using three major search databases including PubMed, Web of Science, and SCOPUS. Search terms included “curcumin derivatives”, “obesity”, and “obesity associated metabolic disease”, and relevant synonyms per search term were used. The search of the literature was conducted from inception until May 2023. Only in vitro and in vivo experimental studies that used synthetic curcumin derivatives with or without pure-form curcumin were selected for further review. Studies assessing curcumin formulations and curcumin metabolites or adjuvants of curcumin and review articles and clinical studies were excluded from this review. Two authors independently conducted the search of the literature and data analysis.

## 2. Physiochemical, Structural, and Biological Properties of Curcumin

Curcumin, also known as diferuloylmethane, is a natural polyphenolic compound originally derived from the rhizome of turmeric (*Curcuma longa* L.) (Figure 1) and other *Curcuma* spp. [16]. Curcumin was first discovered in 1815 [32] and isolated in 1842 [33] by Henri Auguste Vogel and Pierre Joseph Pelletier. Its chemical structure was first characterized as a diferuloylmethane or (1E,6E)-1,7-bis[4-hydroxy-3-methoxyphenyl]-1,6-heptadiene-3,5-dione in 1910 [34], whilst it was first synthesized in 1913 [35]. Curcumin is an orange-yellow crystalline powder, with a chemical formula of C_21_H_20_O_6_, a molecular weight of 368.38 g/mol, and a melting temperature of 183 °C [36]. It is a hydrophobic molecule with a partition coefficient or logP value of approximately 3.0 that is not soluble in water, but only in solvents such as dimethyl sulfoxide, ethanol, methanol, chloroform, acetonitrile, and ethyl acetate [36]. The chemical structure of curcumin is a bis-α,β-unsaturated β-diketone that consists of two aromatic rings, each comprising one hydroxy and one methoxy substituent that are connected by a seven-carbon linker chain consisting of an α,β-unsaturated β-diketone moiety [36,37]. The β-diketone moiety is subject to keto–enol tautomerization with the keto form being predominant under neutral and acidic conditions, while the enol form is predominant in alkaline solutions (Figure 2) [38]. Curcumin, which makes up 60–70% of curcuminoid extract, co-exists with other curcuminoids, including demethoxycurcumin or 1-(4-hydroxy-3-methoxyphenyl)-7-(4-hydroxyphenyl)-1,6-heptadiene-3,5-dione, constituting 20–27% of curcuminoid extract, and bisdemethoxycurcumin or 1,7-bis(4-hydroxyphenyl)-1,6-heptadiene-3,5-dione, constituting 10–15% of curcuminoid extract (Figure 1) [39].

Turmeric has long been used as a traditional medicinal herb to treat various ailments such as wound healing, dermatological diseases, pneumonia, hepatic disorders, coughs, and arthritis [40]. Curcumin itself is known to possess several biological and pharmacological activities including anti-microbial, anti-fungal, anti-viral, antioxidant, anti-cancer, anti-inflammatory, anti-diabetic, and anti-obesity effects, as well as cardioprotective and neuroprotective properties [41]. Many studies have reported that the anti-obesity effects of curcumin are mediated by inhibiting adipogenesis, regulating lipid metabolism, stimulating energy expenditure, and improving gut microbiota as recently reviewed [42]. In addition, curcumin ameliorates obesity-associated metabolic disorders including insulin resistance, inflammation, oxidative stress [43], dyslipidemia, NAFLD [44], CVD [45], T2D [46], and various cancer types [47]. Curcumin has a long-established safety record with a GRAS (generally recognized as safe) status from the Food and Drug Administration [48], and several studies have shown that it is safe to use in animals and humans, even at doses up to 8 g/day [49].

## 3. The Pharmacokinetic Profile of Curcumin

Despite its efficacy and safety, several preclinical and clinical studies have shown that the pharmacokinetic profile of curcumin is unfavorable, leading to its poor bioavailability and its limited clinical applicability [22,23]. The poor bioavailability of curcumin is attributed to its low aqueous solubility and physiological instability. Curcumin has a very low aqueous solubility especially in acidic or neutral pH conditions, while at pH values above neutral it dissociates to increase its solubility but undergoes rapid degradation [50]. The major factors postulated to contribute to curcumin’s low bioavailability include low gastrointestinal absorption, poor distribution, extensive metabolism, and rapid excretion [51]. Following oral administration, the majority of curcumin is excreted unchanged in the feces (≤90%), while only a small proportion is absorbed in the intestine, followed by rapid metabolism in plasma and the liver [51]. Trace amounts of orally ingested curcumin are eliminated as urinary metabolites [52].

The absorption, bioavailability, and distribution of curcumin has been extensively studied in humans and rodents. Preclinical in vivo studies that investigated the pharmacokinetics of curcumin in rats showed that 60–500 ng/mL of curcumin was present in the blood following oral administration of 0.5–1 g curcumin [53,54], indicating that only ~1% of curcumin is bioavailable [54]. Furthermore, a study conducted in healthy human subjects could not detect curcumin in serum following a single oral dose of curcumin standardized powder extract with doses ranging from 0.5 to 12 g [55]. Another study only detected the major conjugate metabolites of curcumin, including curcumin glucuronide and sulfate conjugates but not curcumin in the plasma of healthy human subjects administered either a 10 g or 12 g single dose of curcumin [56]. These studies suggest that curcumin is rapidly metabolized following oral administration and may be detected as metabolites in plasma.

The absorbed curcumin primarily undergoes rapid phase I and phase II metabolism in the hepatic and intestinal tissues [57]. Phase I metabolism occurs due to reduction reactions and leads to the reduction of curcumin to dihydrocurcumin (DHC), tetrahydrocurcumin (THC), hexahydrocurcumin (HHC), and octahydrocurcumin (OHC) by reductases, while phase II metabolism occurs due to conjugation, resulting in curcumin sulfate and curcumin glucuronide (Figure 2) [58]. Phase I reduced metabolites are also transformed into their corresponding glucuronide-conjugates via phase II metabolism [59]. Alternatively, curcumin metabolism also occurs in the intestinal microbiota via a nicotinamide adenine dinucleotide phosphate (NADPH) dependent reductase pathway of *Escherichia coli* to form DHC and THC [60]. Several studies have reported that curcumin metabolites possess various biological activities including antioxidant, anti-inflammatory, anti-cancer, anti-diabetic, cardioprotective, and neuroprotective effects [61]. As such, it has been suggested that the therapeutic efficacy of curcumin is facilitated by its metabolites [58]. Other than the circulation, the levels of curcumin and its metabolites, DHC and THC, were also detected in the livers and kidneys of mice following intravenous administration of curcumin, while only curcumin was found in the brain of these mice [62]. This suggests that curcumin but not its metabolites can cross the blood–brain barrier.

While there is ongoing research on various pharmaceutical strategies to overcome the poor bioavailability of curcumin and subsequently improve its biological efficacy, research efforts to develop curcumin formulations to increase its solubility, improve intestinal stability and absorption route, and inhibit degradation have garnered great interest [51,63]. These formulations include various small particles such as liposomes, micelles, cyclodextrin, microgels, solid lipid nanoparticles, nanostructured lipid carriers, biopolymer nanoparticles, phospholipid complexes, microemulsions, and nano-emulsions that have been complexed with curcumin or used to encapsulate curcumin by means of either chemical or physical assembly (Figure 3) [64]. Although several of these curcumin formulations have been established, only a few have been tested in clinical settings using doses that are clinically relevant when compared with standard turmeric extract containing 95% curcuminoids. For instance, Fança-Berthon and colleagues compared the pharmacokinetics of curcuminoids in five different formulations including a turmeric dried colloidal suspension, a standardized turmeric extract, a liquid micellar preparation, a piperine–curcuminoid combination, and a phytosome formulation and demonstrated that low dose (300 mg) dried colloidal suspension led to high absorption of curcuminoids (both unconjugated and conjugated) compared to a high dose (1500 mg) of standard extract [65].

**Figure 2 ijms-24-14366-f002:**
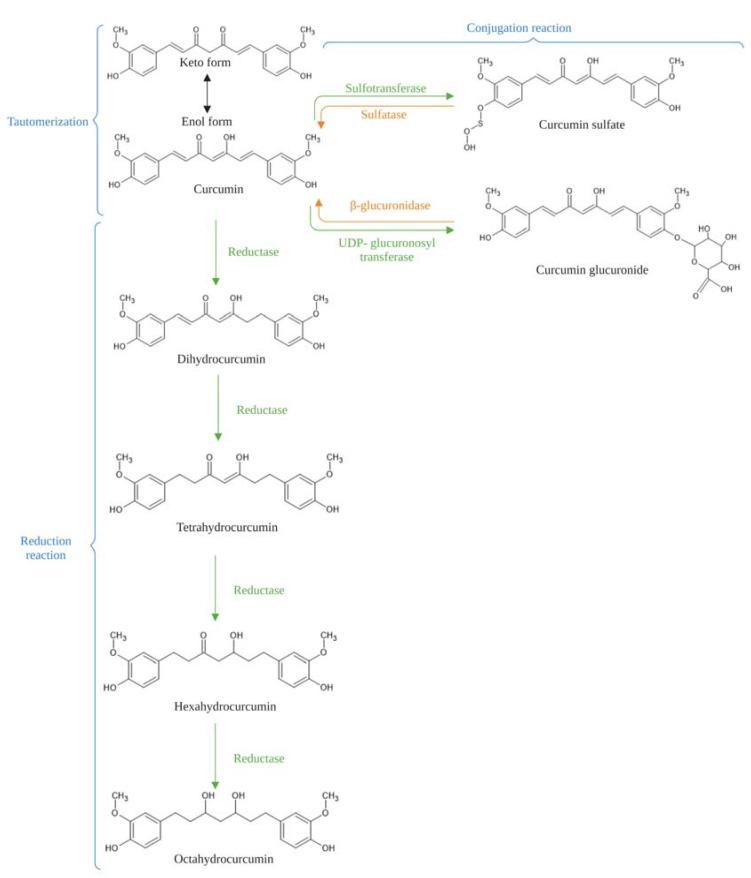
An illustration of curcumin tautomerization and metabolic pathways including the conjugation and reduction reactions. Figure adapted and modified from Pandey et al. [58].

Furthermore, adjunct formulations, also known as combination therapy, produced by complexing curcumin to piperine have been shown to inhibit its rapid metabolism and increase its absorption, thus improving curcumin levels in serum and enhancing its bioavailability (Figure 3) [66]. Other adjuvant formulations such as silibinin, quercetin, and doxorubicin have been explored and shown to improve the biological activity of curcumin (Figure 3) [67,68]. In addition, several studies have intensively explored the use of medicinal chemistry strategies to modify the structure of curcumin by synthesizing curcumin derivatives or analogs, thereby enhancing curcumin’s bioavailability and improving its therapeutic efficacy (Figure 3) [69,70,71]. The pharmacological efficacy of curcumin derivatives is attributed to key elements in its structure [72].

**Figure 3 ijms-24-14366-f003:**
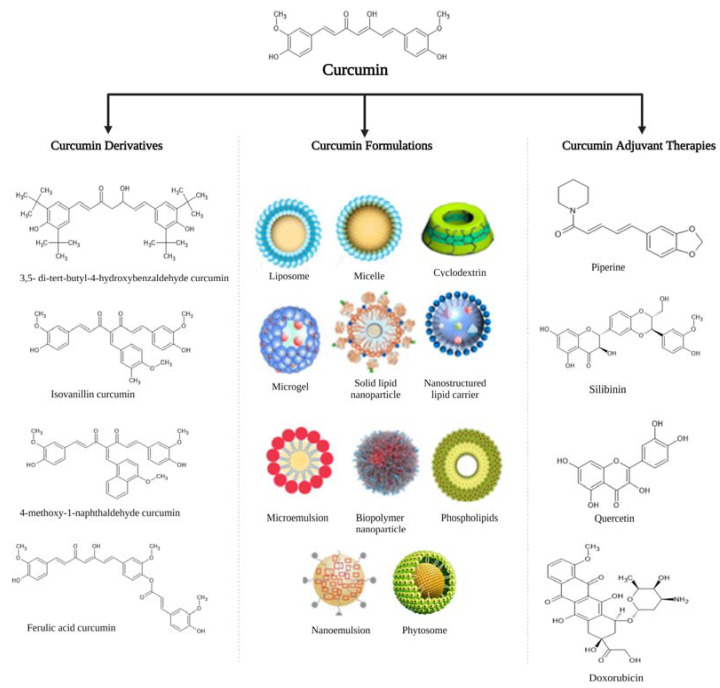
Applications of various strategies (including formulations, adjuvant therapies, and synthetic derivatives) to overcome low bioavailability of curcumin and improve its therapeutic efficacy. Images of the various formulations were obtained from Wong et al. [64] and Pastorelli et al. [73], the structures of adjuvant therapies were taken from Zhang et al. [67] and Moorthi et al. [68], and the structures of curcumin derivatives were obtained from Law et al. [74].

## 4. Efficacy of Curcumin Derivatives against Obesity and Associated Metabolic Complications

In the current review, we identified eight synthetic curcumin derivatives that were shown to ameliorate obesity and associated metabolic complications. All these compounds were reported using in vivo animal models of obesity and associated metabolic complications, while five of these derivatives were also demonstrated in the in vitro cell culture models of obesity and associated metabolic complications. Five of the eight curcumin derivatives reported here were shown to be more effective at ameliorating obesity and its associated metabolic complications when compared to curcumin, while the other three curcumin derivatives were not compared to curcumin.

### 4.1. In Vitro Studies Reporting on the Therapeutic Effects of Curcumin Derivatives against Obesity and Associated Metabolic Complications

Panzhinskiy et al. [75] evaluated a pyrazole derivative of curcumin, CNB-001 or [4-((1E)-2-(5-(4-hydroxy-3-methoxystyryl-)-1-phenyl-1H-pyrazoyl-3-yl)vinyl)-2-methoxy-phenol)] previously synthesized and characterized by Liu and colleagues [76]. It was shown that CNB-001 reversed palmitic acid (PA)-induced insulin resistance by increasing the expression of the phosphorylated form of the insulin receptor (IR), insulin receptor substrate-1 (IRS-1), and protein kinase B (AKT), thereby stimulating insulin-stimulated glucose uptake in C2C12 myotubes (Table 1) [75]. This study did not include curcumin treatment; therefore, it was not possible to assess whether CNB-001 has improved efficacy on insulin resistance compared to curcumin.

Another pyrazole curcumin derivative, Curcumin-3,4-Dichloro Phenyl Pyrazole (CDPP), previously synthesized and shown to exhibit antioxidant, cyclooxygenase inhibitory, and anti-inflammatory effects when using non-cell-based assays [77], was evaluated for its effects on adipogenesis and lipogenesis in vitro in differentiating 3T3-L1 preadipocytes, C3H10T1/2 multipotential mesenchymal stem cells, human mesenchymal stem cells (hMSC), and mature 3T3-L1 adipocytes (Table 1) [78]. CDPP inhibited adipocyte differentiation of these cells by decreasing lipid content, particularly early-phase adipogenesis [78]. CDPP also reduced both messenger ribonucleic acid (mRNA) and protein expression of adipogenesis and lipogenesis-related genes including CCAAT/enhancer-binding protein alpha (C/EBPα), adipocyte Protein 2 (aP2), sterol regulatory element binding transcription factor 1 (SREBP1), peroxisome proliferator-activated receptor gamma (PPARγ), and fatty acid synthase (FAS) [78]. In addition, CDPP was shown to inhibit adipogenesis in 3T3-L1 preadipocytes by suppressing the protein kinase B/mammalian target of rapamycin (AKT/mTOR) pathway and inhibiting mitotic clonal expansion via cell cycle arrest (Table 1) [78]. In mature 3T3-L1 adipocytes, CDPP had no effects on lipolysis but increased oxygen consumption rate and the expression of uncoupling protein 1 (UCP1) and peroxisome proliferator-activated receptor gamma coactivator 1 alpha (PGC1α), suggesting that CDPP increases energy utilization in these cells (Table 1) [78]. Interestingly, when compared to curcumin at 20 μM, this study found that CDPP at 10 and 20 μM is a potent inhibitor of adipogenesis in 3T3-L1 preadipocytes and C3H10T1/2 cells [78].

The therapeutic effect of the curcumin analog, curcumin5-8 (CUR5-8) previously synthesized by Woo et al. [79] was compared with that of curcumin in vitro in PA-induced murine AML12 hepatocyte cells (Table 1) [80]. Lee et al. [80] reported that CUR5-8 ameliorates NAFLD by reducing lipid droplet formation, decreasing SREBP1 expression, and increasing adenosine monophosphate-activated protein kinase (AMPK) phosphorylation. CUR5-8 also induced autophagy by increasing microtubule-associated protein light chain 3-autophagy related 7 (LC3-ATG7) expression and inhibited apoptosis by increasing B-cell lymphoma 2/Bcl-2 associated X protein (BCL-2/BAX) ratio expression in these cells even more effectively than curcumin treatment (Table 1) [80].

Dehydrozingerone (DHZ), (E)-4-(4-hydroxy-3-methoxyphenyl)-but-3-en-2-one, also known as the feruloylmethane half analog of curcumin, is naturally isolated from ginger (*Zingiber officinale*) and also derived through laboratory-based synthesis procedures [81]. Studies have demonstrated the protective effects of DHZ against various metabolic dysfunctions including obesity-associated skeletal muscle insulin resistance [82] and renal lipotoxicity-induced inflammation and reactive oxygen species (ROS) formation [83]. DHZ inhibited inflammatory signals and ROS production by increasing the expression of phosphorylated AMPK and reducing phosphorylated p38 mitogen activated protein kinases (p38 MAPK) expression in high glucose (HG)-induced mouse mesangial (MES-13) cells [83]. Furthermore, the treatment of PA-induced MES-13 cells with DHZ reduced lipid accumulation, decreased phosphorylated p38 MAPK, phosphorylated cAMP response element-binding protein (CREB), and cyclooxygenase 2 (COX2) expression levels (Table 1) [83]. Similarly, in PA exposed mouse podocytes, Lee et al. [83] showed that treatment with DHZ reduced ROS formation and exerted protective effects against cellular toxicity by regulating oxidative stress and lipid metabolism (Table 1). In C2C12 myoblasts, DHZ activated AMPK phosphorylation and the p38 MAPK signaling pathway, upregulated glucose transporter type 4 (GLUT4) mRNA and protein expression, and increased IRS-1 and AKT phosphorylation, while in L6 myotubes DHZ improved insulin resistance by stimulating basal and insulin-stimulated glucose uptake [82]. Furthermore, these in vitro experiments confirmed that DHZ exhibited stronger effects for AMPK phosphorylation than curcumin [82,83].

Over the years, several monocarbonyl analogs of curcumin, including (2E,6E)-2,6-bis[2-(trifluoromethyl)benzylidene]cyclohexanone (the so-called C66 curcumin analog), have been widely synthesized and studied for their biological activities [84,85,86,87], including their potential to modulate obesity-associated metabolic complications such as obesity-induced renal injury [88] and myocardial dysfunction [89]. In cultured mouse glomerular mesangial cells (SV40-MES-13), C66 was able to attenuate the profibrotic mechanisms, apoptosis, inflammatory responses, and the nuclear factor kappa B/c-Jun N-terminal kinase (NF-κB/JNK) activated signaling pathway induced by treatment with PA (Table 1) [88]. Similarly, in PA-challenged H9c2 cardiomyocytes, C66 provided cardio-protection by preventing myocardial injury and apoptosis, which was accompanied by inhibition of the NF-κB/JNK activation pathway and reduced inflammation (Table 1) [89]. Although these studies investigated the ameliorative effects of C66 in obesity-induced renal and myocardial injuries, none of them compared the efficacy of C66 with curcumin.

**Table 1 ijms-24-14366-t001:** Summary of in vitro studies reporting on the therapeutic effects of curcumin derivatives on obesity and associated metabolic complications.

Curcumin Derivative	Chemical Structure	Aim, Experimental Model, Treatment Dose, and Treatment Period	Outcomes	Reference
CNB-001 [4-((1E)-2-(5-(4-hydroxy-3-methoxystyryl-)-1- phenyl-1H-pyrazoyl-3-yl) vinyl)-2-methoxy-phenol]	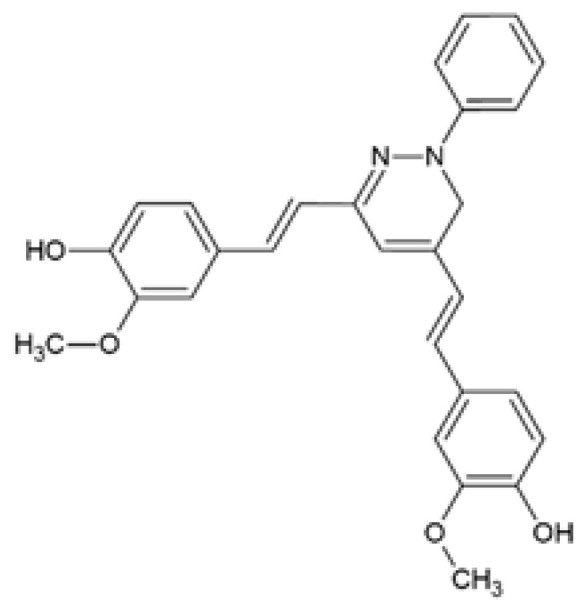	The effect of CNB-001 on insulin resistance was examined. Mouse C2C12 myotubes were co-treated with 400 µM palmitic acid (PA) and 0.1, 1, or 5 μM CNB-001 for 12 h under basal or insulin-stimulated conditions (50 nM insulin, 30 min).	CNB-001 restored PA-induced insulin resistance and impaired insulin signaling by improving insulin-stimulated glucose uptake and increasing insulin-stimulated p-IR, p-IRS-1, and p-AKT expression levels.	[75]
Curcumin-3,4-Dichloro Phenyl Pyrazole (CDPP) 4,4′-(1E,1′E)-2,2′-(1-(3,4-dichlorophenyl)-1H-pyrazole-3,5- diyl) bis(ethene-2,1-diyl) bis(2-methoxyphenol)	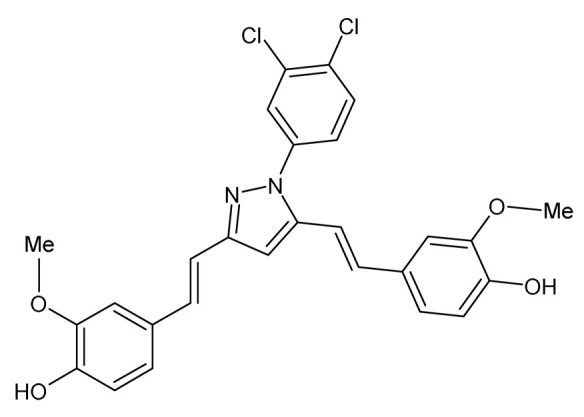	The effects of CDPP on adipogenesis and lipogenesis were investigated. Mouse 3T3-L1 preadipocytes, mouse C3H10T1/2 mesenchymal stem cells, and human mesenchymal stem cells (hMSC) were treated with 5, 10, and 20 μM CDPP or 20 μM curcumin during differentiation (7 days). Mature 3T3-L1 adipocytes were treated with 20 μM CDPP for 24 h.	CDPP inhibited adipogenesis by dose-dependently decreasing lipid accumulation, reducing C/EBPα, aP2, SREBP1c, PPARγ, and FAS mRNA and protein expression, and downregulating the AKT/mTOR pathway. In 3T3-L1 preadipocytes, CDPP inhibited mitotic clonal expansion by arresting cells at G1-phase and S-phase, decreasing Cyclin D1, Cyclin D3, CDK2, CDK4, CDK6, and p-ERK1/2 expression, and increasing p-p27 expression. CDPP increased oxygen consumption rate and UCP1 and PGC1α expression in 3T3-L1 adipocytes. Compared to curcumin, adipogenesis was significantly inhibited by CDPP.	[78]
Curcumin5-8 (CUR5-8)	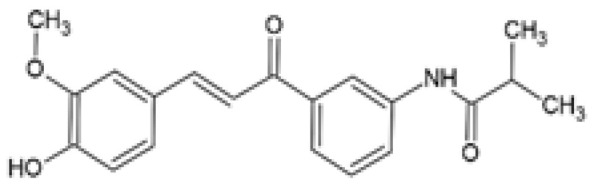	The effect of CUR5-8 on hepatic steatosis was investigated. Mouse AML12 hepatocyte cells were treated with 250 μM PA with or without 10 μM CUR5-8 or curcumin for 24 h.	CUR5-8 decreased PA-induced lipid droplet formation by reducing SREBP1 expression and increasing p-AMPK. CUR5-8 induced autophagy via increased expression of LC3-ATG7, and inhibited apoptosis via increased BCL-2/BAX ratio. The efficacy of CUR5-8 on increased AMPK phosphorylation and autophagy formation and reduced apoptosis was significant compared to curcumin.	[80]
Dehydrozingerone (DHZ) Feruloylmethane 1, 7-bis (4-hydroxy-3-methoxyphenyl)-1, 6-heptadiene-3, 5-dione	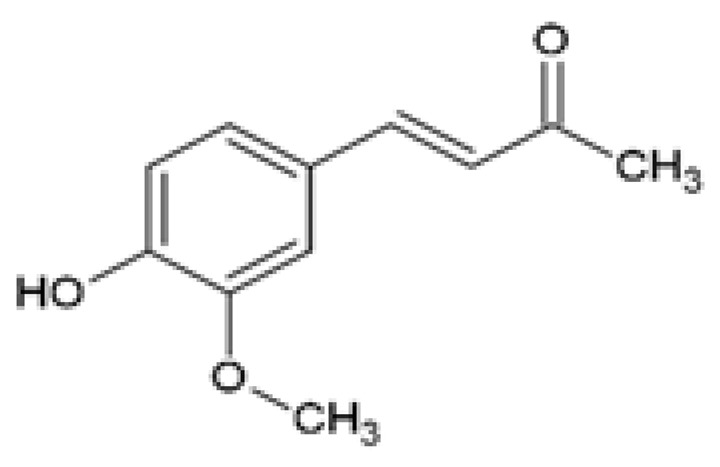	The efficacy of DHZ was evaluated on renal lipotoxicity. Differentiated mouse podocytes were treated with 400 µM PA in the presence or absence of 20 µM DHZ for 24 h.	DHZ inhibited ROS formation, reduced protein expression of NOX-4, NRF2, HO-1, BAX, and SREBP1, and downregulated mRNA expression of *Tgf-β* and *Icam*. DHZ increased BCL-2 and podocin protein expression.	[83]
		The efficacy of DHZ was evaluated on renal lipotoxicity. Mouse glomerular mesangial MES-13 cells were treated with either 30 mM of high glucose (HG) or 250 μM PA in the presence or absence of 20 µM DHZ for 24 h. MES-13 cells were treated with 20 µM DHZ or 20 µM curcumin for 24 h.	DHZ reduced HG-induced ROS production and inflammatory signals by increasing p-AMPK and reducing p-p38 MAPK expression. DHZ decreased PA-induced lipid accumulation, decreased p-p38 MAPK, p-CREB, and COX2 expression, and increased p-AMPK activation. The p-AMPK expression was significantly higher in DHZ-treated cells compared with curcumin.	
		The effect of DHZ on the metabolic profile of skeletal muscle was evaluated. Mouse C2C12 myoblasts were treated with 1, 3, 10, 30, or 100 µM DHZ for either 0.1, 0.5, 1, 3, 6, 12, or 24 h. Rat L6 skeletal muscle myotubes were treated with 1, 3, 10, or 30 µM DHZ for 1 h in the presence or absence of 100 nM insulin.	DHZ increased the activation of p-AMPK and p-p38 MAPK signal pathways in C2C12 cells. DHZ upregulated GLUT4, p-IRS-1, and p-AKT expression in C2C12 cells. DHZ induced basal and insulin-stimulated glucose uptake in L6 myotubes. Compared with curcumin, p-AMPK expression was significantly higher in DHZ treated C2C12 cells.	[82]
C66 (2E,6E)-2,6-bis (2-(tri- fluoromethyl)benzylidene) cyclohexanone	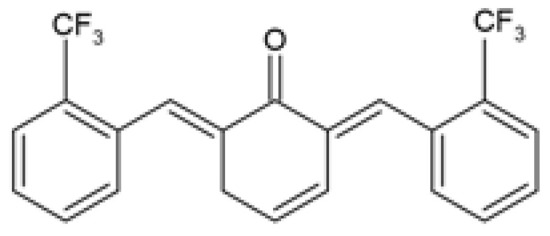	The protective effect of C66 on renal lipotoxicity and injury was investigated. Mouse glomerular mesangial SV40-MES-13 cells were pre-treated with 2.5, 5, and 10 μM C66 for 2 h followed by treatment with 200 μM PA for 2, 12, or 24 h.	C66 attenuated PA-induced fibrosis by reducing mRNA and protein expression of TGF-β, Col-IV, and MMP-9. C66 ameliorated PA-induced apoptosis by reducing apoptotic positive cells, decreasing BAX and cleaved caspase-3 expression, and increasing BCL-2 expression. C66 protected against PA-induced inflammation by reducing NF-κB/JNK activation via increased IκBα accumulation, reduced p65 phosphorylation and translocation to the nucleus, and reduced JNK phosphorylation. C66 reduced TNF-α, IL-6, and IL-1β secretion levels and mRNA expression.	[88]
		The protective effect of C66 on cardiomyocyte dysfunction was tested. Rat embryonic heart-derived H9c2 cells were pre-treated with 2.5, 5, and 10 μΜ C66 for 2 h, then treated with 200 μM PA for 24 h.	C66 prevented PA-induced hypertrophy and fibrosis by reducing MyHC, ANP, BNP, TGF-β, Col-I, and MMP-9 mRNA and protein expression. C66 reduced PA-induced apoptosis by decreasing positive apoptotic cells, reducing expression of cleaved caspase-3 and BAX, and increasing BCL-2 expression. C66 ameliorated PA-induced inflammation by reducing NF-κB activation via reduced p65 phosphorylation and translocation to the nucleus and increased IκBα accumulation. C66 also reduced JNK activation and decreased *Tnf-α*, *Il-6*, *Il-1β*, *Vcam-1*, and *Icam-1* mRNA expression.	[89]

Abbreviations: ANP: A-type natriuretic peptide; aP2: adipocyte Protein 2; ATG7: autophagy related 7; BAX: Bcl-2 associated X protein; BCL-2: B-cell lymphoma 2; BNP: B-type natriuretic peptide; C/EBPα: CCAAT/enhancer-binding protein alpha; CDK2/4/6: cyclin-dependent kinase 2/4/6; CDPP: Curcumin-3:4-Dichloro Phenyl Pyrazole; Col-I: collagen type-I; Col-IV: collagen type-IV; COX2: cyclooxygenase 2; CUR5-8: Curcumin5-8; DHZ: Dehydrozingerone; FAS: fatty acid synthase; GLUT4: glucose transporter type 4; HG: high glucose; hMSC: human mesenchymal stem cell; HO-1: heme oxygenase-1; ICAM-1: intercellular adhesion molecule-1; IL-1β: interleukin 1 beta; IL-6: interleukin 6; IκBα: inhibitor of nuclear factor kappa B; JNK: c-Jun N-terminal kinase; LC3: light chain 3; MMP-9: matrix Metallopeptidase 9; mRNA: messenger ribonucleic acid; mTOR: mammalian target of rapamycin; MyHC: myosin heavy chain; NF-κB: nuclear factor kappa B; NOX-4: NADPH oxidase-4; NRF2: nuclear factor erythroid 2–related factor 2; p38 MAPK: p38 mitogen-activated protein kinases; p65: nuclear factor kappa B subunit; PA: palmitic acid; p-AKT: phosphorylated protein kinase B; p-AMPK: phosphorylated adenosine monophosphate-activated protein kinase; p-CREB: phosphorylated cAMP response element-binding protein; p-ERK1/2: phosphorylated extracellular signal-regulated protein kinases 1 and 2; PGC1α: peroxisome proliferator-activated receptor gamma coactivator 1 alpha; p-IR: phosphorylated insulin receptor; p-IRS-1: phosphorylated insulin receptor substrate-1; PPARγ: peroxisome proliferator-activated receptor gamma; ROS: reactive oxygen species; SREBP1: sterol regulatory element binding transcription factor 1; TGF-β: transforming growth factor beta; TNF-α: tumor necrosis factor alpha; UCP1: uncoupling protein 1; VCAM-1: vascular cell adhesion molecule-1.

### 4.2. In Vivo Studies Reporting on the Therapeutic Effects of Curcumin Derivatives on Obesity-Associated Metabolic Complications

An ortho-hydroxyl group-substituted analog of curcumin (known as salicylcurcumin), previously synthesized [90] and reported to be effective in treating skin tumors compared to other curcuminoids such as curcumin [91], was evaluated for its efficacy to ameliorate alcohol and polyunsaturated fatty acids (PUFAs)-induced hyperlipidemia when compared to curcumin [92]. It was shown that salicylcurcumin reduced cholesterol, triglycerides (TGs), free fatty acids (FFAs), and phospholipids (PLs) plasma levels (Table 2) [92]. Furthermore, salicylcurcumin treatment was demonstrated to be more effective in attenuating the circulating lipid profiles than curcumin in this study [93].

In high fat diet (HFD)-fed mice, CNB-001 treatment reversed obesity-associated metabolic complications by reducing body weight gain, decreasing overall adiposity, and increasing energy expenditure (Table 2) [75]. In addition, CNB-001 ameliorated hepatic steatosis while decreasing serum triglyceride and interleukin 6 (IL-6) levels. Furthermore, CNB-001 improved glucose tolerance and insulin resistance and increased ex vivo gastrocnemius muscle glucose uptake (Table 2) [75]. At the molecular level, CNB-001 restored insulin signaling by inducing the phosphorylation of AKT and IR, attenuated endoplasmic reticulum (ER) stress by decreasing eukaryotic initiation factor 2 alpha (eIF2α) phosphorylation and glucose-regulated protein 78 (GRP78) expression, and reduced protein tyrosine phosphatase 1B (PTP1B) expression in the gastrocnemius muscles of HFD-fed mice (Table 2) [75]. While CNB-001 treatment had no significant effect in the low-fat diet (LFD)-fed mice, this study did not assess whether CNB-001 has improved efficacy compared to curcumin [75].

Another novel monocarbonyl curcumin analog, (2E,6E)-2-(2-bromobenzylidene)-6-(2-(trifluoromethyl) benzylidene) cyclohexanone (Y20), was designed and synthesized by Qian and colleagues via structural modification of the C66 curcumin analog by substituting one of the trifluoromethyls with a bromine [94]. Y20 was hypothesized to possess dual anti-inflammatory and antioxidant properties, hence being investigated for its cardioprotective effects in obese rats induced by HFD [94]. It was reported that Y20 attenuated HFD-induced myocardial inflammation by reducing the expression of cardiac pro-inflammatory markers including tumor necrosis factor alpha (TNF-α), cluster of differentiation 68 (CD68), *Il-6*, interleukin 1 beta (*Il-1β*), and cyclooxygenase-2 (*Cox-2*) (Table 2) [94]. Y20 also ameliorated HFD-induced myocardial inflammation by reducing the expression of cell adhesion molecules including vascular cell adhesion molecule-1 (VCAM-1) and intercellular adhesion molecule-1 (*Icam-1*) while reversing the degradation of inhibitor of nuclear factor kappa B (IκB) (Table 2) [94].

The administration of Y20 reversed HFD-induced cardiac oxidative stress by reducing superoxide anion production and 3-nitrotyrosine (3-NT) accumulation and increasing the expression of nuclear factor erythroid 2–related factor 2 (NRF2) and its downstream anti-oxidant genes, including heme oxygenase-1 (*Ho-1*), and NADPH quinone dehydrogenase 1 (*Nqo-1*) (Table 2) [94]. Moreover, these authors suggested that the attenuation of myocardial inflammation and oxidative stress by Y20 treatment subsequently improved cardiac remodeling by reducing hypertrophy, fibrosis, and apoptosis, although decreased body weight gain and serum TGs levels may also partially contribute to the cardioprotective effects of Y20 (Table 2) [94]. Interestingly, data from this study demonstrated that Y20 exhibited more anti-inflammatory, antioxidant, anti-fibrosis, and anti-apoptotic effects than curcumin, even at a lower dose (20 mg/kg for Y20 vs. 50 mg/kg for curcumin) [94].

Similarly to the in vitro data, Gupta and et al. also demonstrated the ameliorative effects of the curcumin derivative CDPP on obesity and dyslipidemia in an HFD-fed Syrian golden hamster model [78]. This was mediated by significant reductions in body weight gain and reduced serum TGs, total cholesterol (TC), and low-density lipoprotein cholesterol (LDL-c) levels, as well as decreased TGs/high-density lipoprotein cholesterol (HDL-c) ratio (Table 2) [78]. The serum aspartate aminotransferase (AST) and alanine transaminase (ALT) levels were also significantly reduced, suggesting the hepatoprotective effects of CDPP (Table 2) [78]. CDPP also reduced liver and epididymal white adipose tissue (eWAT) weights, decreased hepatic lipid accumulation and adipocyte hypertrophy, and lowered expression of FAS, C/EBPα, aP2, and PPARγ as well as their translated proteins in eWAT, while upregulating hepatic mRNA levels of peroxisome proliferator-activated receptor alpha (*Pparα*) and liver X receptor alpha (*Lxrα*) (Table 2) [78]. These authors also reported that CDPP attenuated dyslipidemia by activating the reverse cholesterol transport from adipose tissue to liver via related gene expression (Table 2). Cumulatively, the results from this study show that CDPP was more effective in ameliorating obesity-induced dyslipidemia than curcumin [78].

The structural modification of natural compounds such as curcumin with polyethylene glycol (PEG) has been shown to improve solubility and physicochemical properties of curcumin [95]. As such, a water-soluble curcumin derivative was synthesized by conjugating two low molecular weight PEGs (mPEG454) via β-thioester bonds to curcumin (Curc-mPEG454) and was shown to improve water solubility (>50 mg/mL) and produce approximately 50–500 times higher serum curcumin levels and greater curcumin distribution in the liver tissue compared to curcumin [96]. Moreover, Curc-mPEG454 previously displayed potent anti-inflammatory [30], anti-fibrotic [97], anti-oxidant [30], anti-steatosis [95] effects. Of interest in the current review is the hepatic lipid lowering effect of Curc-mPEG454 in NAFLD-induced obese mice as previously reported [98]. These authors showed that treatment with Curc-mPEG454 reduced body weight, serum and liver TGs levels, and hepatic lipid accumulation in HFD-induced NAFLD mice (Table 2) [98]. Moreover, Curc-mPEG454 reversed HFD-induced hepatic steatosis and hypertriglyceridemia by modulating the hepatic CREB/PPARγ/cluster of differentiation 36 (CD36) pathway through the activation of CREB phosphorylation and inhibition of PPARγ and CD36 expression (Table 2) [98]. However, these authors did not assess whether Curc-mPEG454 has improved efficacy compared to curcumin.

Like the in vitro data, Lee et al. [80] also reported ameliorative effects of CUR5-8 against NAFLD and insulin resistance in HFD-induced obese mice. In this study, treatment with CUR5-8 was shown to decrease bodyweight gain. CUR5-8 improved insulin resistance by restoring insulin sensitivity, reducing insulin levels, and lowering homeostasis model assessment-estimated insulin resistance (HOMA-IR) values (Table 2) [80]. In addition, CUR5-8 blocked hepatic lipogenesis and TGs accumulation by regulating the expression of lipogenic genes such as decreased FAS, SREBP1, Adipophilin, and PPARγ expression and increased phosphorylation of AMPK (Table 2) [80]. CUR5-8 also inhibited apoptosis by regulating autophagy-related genes such as decreased autophagy related 5 (ATG5) and increased sequestosome 1 (p62) expression (Table 2) [80]. CUR5-8 was shown to reduce body weight gain, enhance insulin sensitivity, and decrease fatty liver to a greater extent than curcumin in this study [80].

The efficacy of DHZ to ameliorate insulin resistance and renal lipotoxicity in vivo in animal models of obesity was also investigated [82,83]. In HFD-fed mice, DHZ suppressed body weight gain and visceral fat accumulation while also reversing hyperglycemia by lowering plasma glucose, insulin, and leptin levels and improving glucose tolerance (Table 2) [82]. Treatment with DHZ also reduced liver weight and fatty liver accumulation, and this was mediated by a reduction in the hepatic expression of gluconeogenic genes, including phosphoenolpyruvate carboxykinase (*Pepck*) and glucose-6-phosphatase (*G6Pase*), and the insulin resistance marker, *Fetuin* (Table 2) [82]. A follow-up study by Lee et al. [83] reported that DHZ decreased body weight gain and reversed HFD-induced renal damage by suppressing inflammation (Table 1). DHZ also ameliorated renal lipid metabolism dysregulated by HFD (Table 2) [83]. These studies did not compare the efficacy of DHZ to curcumin in their in vivo models.

In addition to the in vitro studies, the protective effect of C66 on renal dysfunction [88] and myocardial injury [89] was demonstrated in vivo in HFD-induced obese mice. Along with reducing body weight gain, decreasing serum TGs, LDL-c, and cholesterol levels, and increasing serum HDL-c levels, treatment with C66 attenuated renal dysfunction and adverse kidney remodeling by improving renal function and suppressing kidney fibrosis (Table 2) [88]. In addition, C66 treatment ameliorated HFD-induced renal apoptosis and chronic inflammation via inhibiting NF-κB/JNK activation (Table 2) [88]. Likewise, treatment of HFD-induced obese mice with C66 decreased body weight, reduced serum levels of TGs, LDL-c, and cholesterol and increased serum HDL-c levels, while exhibiting protective effects against myocardial dysfunction and hypertrophy (Table 2) [89]. These authors similarly reported that C66 treatment ameliorated HFD-induced cardiac fibrosis, apoptosis, and inflammation via the inhibition of the NF-κB/JNK activation pathway (Table 2) [89]. The efficacy of C66 was not compared to curcumin in these studies.

**Table 2 ijms-24-14366-t002:** Summary of in vivo studies reporting on the therapeutic effect of curcumin derivatives on obesity and associated metabolic complications.

Curcumin Derivative	Chemical Structure	Aim, Experimental Model, Treatment Dose, and Treatment Period	Outcomes	Reference
Salicylcurcumin *O*-hydroxy-substituted analog of curcumin	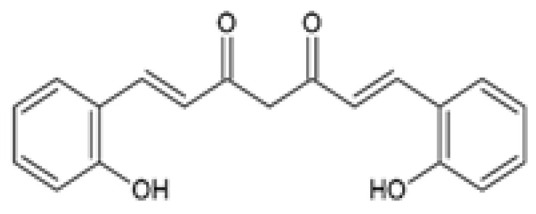	The antihyperlipidemic effect of salicylcurcumin was investigated. Male albino wistar rats were either orally administered 20% EtOH (alcohol-induced) or fed 15% sunflower oil in a HFD (ΔPUFA-induced) or fed a combination of both in the presence or absence of 80 mg/kg salicylcurcumin or curcumin for 45 days.	Salicylcurcumin reduced TC, TGs, FFAs, and PLs plasma levels. The reduction in TC, TGs, FFAs, and PLs plasma levels was more significant in the salicylcurcumin-treated groups compared to the curcumin-treated groups.	[93]
CNB-001 [4-((1E)-2-(5-(4-hydroxy-3-methoxystyryl-)-1- phenyl-1H-pyrazoyl-3-yl) vinyl)-2-methoxy-phenol]	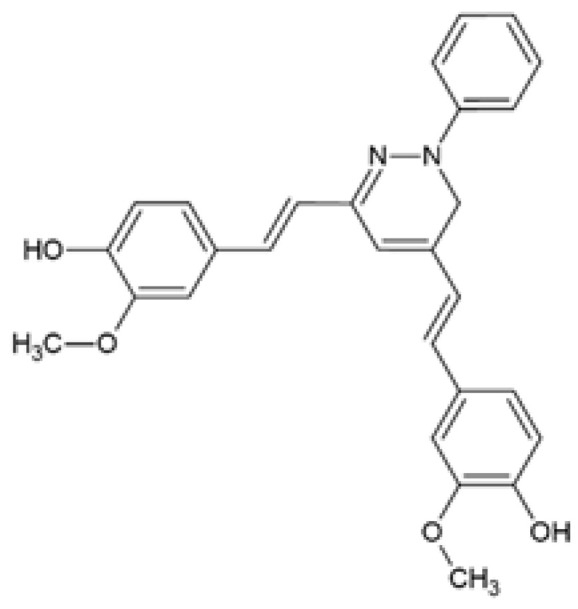	The effect of CNB-001 on obesity-induced insulin resistance was examined. Male C57BL6 mice were fed a HFD and intraperitoneally injected with CNB-001 (40 mg/kg/day, except weekends) for 22 weeks.	CNB-001 reduced BW gain, adiposity, heart weight, liver weight, serum TGs and IL-6 levels. CNB-001 increased energy expenditure, reduced fasting glucose concentrations, improved glucose tolerance and insulin sensitivity, and increased ex vivo gastrocnemius muscle glucose uptake. CNB-001 attenuated hepatic steatosis by decreasing lipid accumulation and hepatic TG content. CNB-001 restored insulin signaling via upregulation of p-AKT and p-IR expression, reduced ER stress via decreased p-eIF2α and GRP78 expression, and decreased PTP1B expression in gastrocnemius muscles.	[75]
Y20 (2E,6E)-2-(2-bromobenzylidene)-6-(2(trifluoromethyl) benzylidene) cyclohexanone	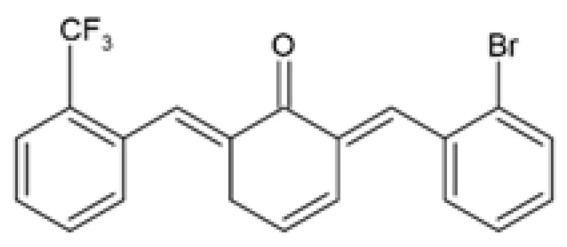	The protective effects and underlying mechanisms of Y20 on obesity-induced cardiac injury was investigated. HFD-fed male Wistar rats were orally gavaged with 20 mg/kg Y20 or 50 mg/kg curcumin daily for 4 weeks during HFD feeding.	Y20 reduced BW gain and serum TGs levels. Y20 attenuated myocardial inflammation by decreasing the protein and mRNA expression of TNF-α, CD68, IL-6, IL-1β, COX-2, VCAM-1, and ICAM-1, and reversing IκB degradation. Y20 reversed cardiac oxidative stress by reducing superoxide anion production and 3-NT accumulation and increasing the protein and mRNA expression of NRF2 and its downstream antioxidant genes, *Ho-1* and *Nqo-1*. Y20 attenuated cardiac hypertrophy by decreasing cardiomyocyte size and the protein and mRNA expression of cardiac hypertrophic markers, ANP and BNP. Y20 ameliorated cardiac fibrosis by reducing Col-I synthesis and deposition and decreasing the protein and mRNA expression of Col-1, TGF-β, MMP-2, and MMP-9. Y20 reversed myocardial apoptosis by reducing the percentage of apoptotic cardiac cells, increasing BCL-2 expression, and decreasing BAX and cleaved PARP expression levels. Y20 exhibited more anti-inflammatory, antioxidant, anti-fibrosis, and anti-apoptotic efficacy than curcumin.	[94]
Curcumin-3,4-Dichloro Phenyl Pyrazole (CDPP) 4,4′-(1E,1′E)-2,2′-(1-(3,4-dichlorophenyl)-1H-pyrazole-3,5- diyl) bis(ethene-2,1-diyl) bis(2-methoxyphenol)	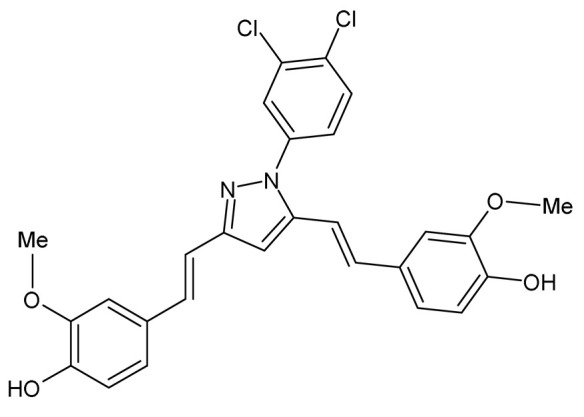	The role of CDPP was evaluated on obesity-induced dyslipidemia. Male Syrian golden hamsters were fed a HFD for 4 days then orally administered either 100 mg/kg CDPP or curcumin daily for 7 days during HFD feeding.	CDPP reduced BW gain and serum levels of TGs, TC, LDL-c, ALT, AST, and the TGs/HDL-c ratio. CDPP decreased serum AST and ALT levels. CDPP reduced liver weight, decreased hepatic lipid accumulation, and upregulated hepatic *Pparα* and *Lxrα* mRNA expression. CDPP decreased eWAT weight, adipocyte hypertrophy, and reduced the expression of FAS, C/EBPα, aP2 and PPARγ in eWAT. CDPP activated reverse cholesterol transport machinery from adipose tissue to liver by increasing *Abca1* and *Srb1* and decreasing *Abcg1* mRNA expression in eWAT, and increasing hepatic *Abcg8*, *Lcat*, *Cyp7a1*, and *Srb1* mRNA expression. The efficacy of CDPP in ameliorating dyslipidemia was more significant compared to curcumin.	[78]
Curc-mPEG454 PEGylated curcumin derivative	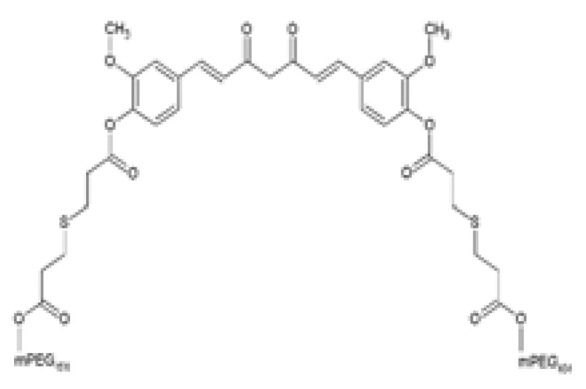	The effects of Curc-mPEG454 on obesity-associated hepatic steatosis was evaluated. HFD-fed male C57BL/6J mice were intraperitoneally injected every other day with 50 and 100 mg/kg Curc-mPEG454 for 16 weeks during HFD feeding.	Curc-mPEG45 lowered BW gain and serum TGs levels. Curc-mPEG454 reduced hepatic steatosis by decreasing TGs content, decreasing lipid accumulation, and attenuating macrovesicular and microvesicular steatosis. Curc-mPEG454 reduced hepatic PPAR𝛾 and CD36 protein and mRNA expression. Curc-mPEG454 activated hepatic p-CREB.	[98]
Curcumin5-8 (CUR5-8)	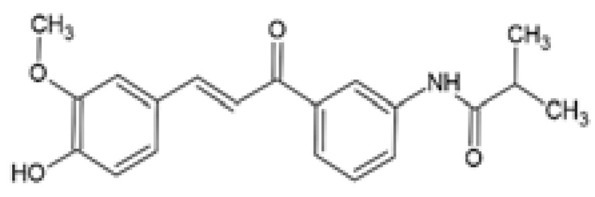	The effect of CUR5-8 on obesity-associated hepatic steatosis and insulin resistance was investigated. Male C57BL/6 mice were fed a HFD chow mixed with either CUR5-8 or curcumin (100 mg/kg/day) for 13 weeks.	CUR5-8 reduced BW gain without suppressing appetite. CUR5 reversed insulin resistance by improving insulin sensitivity, lowering serum insulin levels, and reducing the HOMA-IR index. CUR5-8 protected the liver by decreasing liver weight, reducing fatty liver and TG accumulation, and reducing serum levels of ALT. CUR5-8 regulated hepatic lipid metabolism by decreasing FAS, SREBP1, adipophilin, and PPARγ mRNA and protein expression and increasing p-AMPK expression. CUR5-8 regulated autophagy in the liver by decreasing ATG5 and increasing p62 expression. CUR5-8 reduced apoptosis in the liver by decreasing cleaved caspase-3 staining. Compared to curcumin, the efficacy of CUR5-8 in ameliorating hepatic steatosis and insulin resistance was more significant.	[80]
Dehydrozingerone (DHZ) Feruloylmethane 1, 7-bis (4-hydroxy-3-methoxyphenyl)-1, 6-heptadiene-3, 5-dione	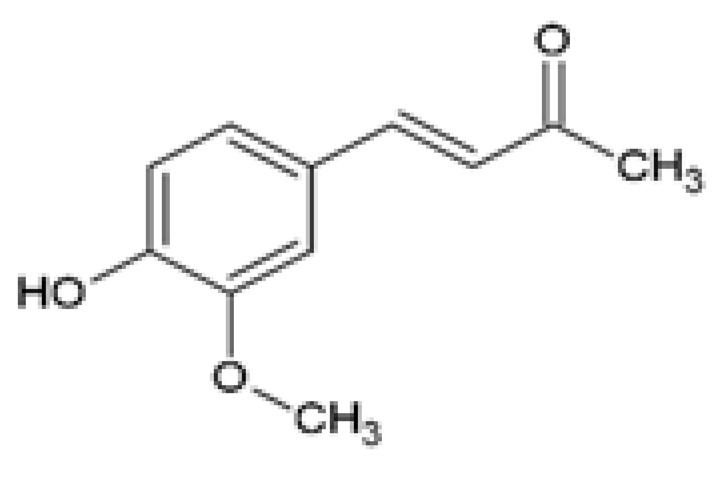	The efficacy of DHZ on obesity-induced renal lipotoxicity, skeletal muscle metabolism, and the underlying mechanisms was investigated. Male C57BL/6 mice were fed a HFD with or without 100 mg/kg DHZ for 12 weeks.	DHZ reduced BW gain and restored HFD-induced renal damage by decreasing kidney weight, urinary albumin, and urinary ACR. DHZ increased *Podocin* and decreased *Il-1β* mRNA expression. DHZ increased nephrin expression and reduced CD68 and ARGINASE 2 expression. DHZ regulated renal lipid metabolism by reducing renal glycerol, FFAs and cholesteryl ester levels, decreasing mRNA and protein expression of SREBP1, SREBP2, FAS, and ACC, and increasing mRNA and protein expression of PPARα, CPT1, and p-AMPK.	[83]
			DHZ suppressed HFD-induced BW gain, decreased peri-renal fat, epididymal fat, and liver weight, and reduced adipocyte size and lipid accumulation in the liver and epididymal fat. DHZ decreased plasma glucose, insulin, and leptin levels, and improved glucose tolerance. DHZ decreased hepatic mRNA expression of *G6Pase*, *Pepck*, and *Fetuin*.	[82]
C66 (2E,6E)-2,6-bis (2-(tri- fluoromethyl)benzylidene) cyclohexanone	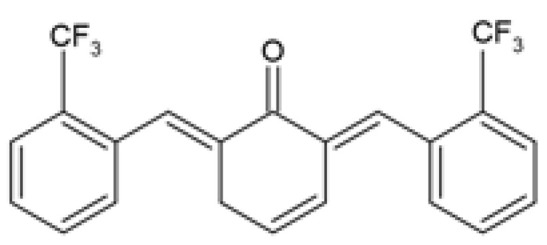	The protective effect of C66 on obesity-induced renal dysfunction was evaluated. HFD-fed SPF C57BL/6 male mice were orally administered 5, 10, and 20 mg/kg of C66 every 2 days for 11 weeks during HFD feeding.	C66 reduced BW gain, decreased serum TGs and LDL-c levels, and increased serum HDL-c levels. C66 decreased kidney weight, serum BUN and CRE levels, and urine CRE and albumin levels. C66 reduced fibrosis by reducing collagen and ECM deposition and decreasing the mRNA and protein expression of Col-IV, TGF-β, and MMP-9. C66 reversed apoptosis by reducing TUNEL positive stained cells, decreasing cleaved caspase-3 and BAX expression, and increasing BCL-2 expression. C66 attenuated chronic inflammation by reducing macrophage infiltration, decreasing *Tnf-α*, *Il-6*, and *Il-1β* mRNA expression levels, and inhibiting NF-κB/JNK activation via increased IκBα accumulation, and reduced expression of p-p65 and p-JNK.	[88]
		The protective effect of C66 on obesity-induced cardiomyopathy through inhibition of JNK-mediated inflammation was investigated. Male C57BL/6 mice were fed HFD for 11 weeks followed by oral administration with 5, 10, and 20 mg/kg of C66 every 2 days for a further 7 weeks during HFD feeding.	C66 reduced BW gain and decreased serum TGs, LDL-c, and cholesterol levels, and increased HDL-c serum levels. C66 restored cardiac function by improving EF% and FS%. C66 decreased myocardial hypertrophy by reducing heart weight to tibia length ratio, cardiomyocyte longitudinal and transverse section area, and the circulating and mRNA expression levels of CK-MB, ANP, and BNP. C66 attenuated cardiac fibrosis by decreasing collagen deposition and reducing MyHc, TGF-β, and Col-I protein and mRNA expression levels. C66 suppressed myocardial apoptosis by reducing TUNEL positive stained cells, decreasing cleaved caspase-3 and BAX expression, and increasing BCL-2 expression. C66 ameliorated cardiac inflammation by decreasing macrophage infiltration, reducing *Tnf-α*, *Il-6*, and *Il-1β* mRNA expression levels, and reversing NF-κB/JNK activation via increased IκBα accumulation, and reduced expression of p-p65 and p-JNK.	[89]

Abbreviations: 3-NT: 3-nitrotyrosine; ABCG1: ATP binding cassette subfamily G member 1; ABCG8: ATP binding cassette subfamily G member 8; ABCA1: ATP binding cassette subfamily A member 1; ACC: acetyl-CoA carboxylase; ALT: alanine transaminase; ANP: A-type natriuretic peptide; aP2: adipocyte Protein 2; AST: aspartate aminotransferase; ATG5: autophagy related 5; BAX: Bcl-2 associated X protein; BCL-2: B-cell lymphoma 2 protein; BNP: B-type natriuretic peptide; BUN: blood urea nitrogen; BW: body weight; C/EBPα: CCAAT/enhancer-binding protein alpha; CD36: cluster of differentiation 36; CD68: cluster of differentiation 68; CDPP: Curcumin-3:4-Dichloro Phenyl Pyrazole; CK-MB: creatine kinase MB; Col-I: collagen type-I; Col-IV: collagen type-IV; COX-2: cyclooxygenase-2; CPT1: carnitine palmitoyltransferase 1; CRE: creatinine; CUR5-8: Curcumin5-8; Curc-mPEG454: PEGylated curcumin derivative; CYP7A1: cholesterol 7 alpha-hydroxylase; DHZ: Dehydrozingerone; ECM: extracellular matrix; EF%: percentage of ejection fraction; ER: endoplasmic reticulum; EtOH: ethanol; eWAT: epididymal white adipose tissue; FAS: fatty acid synthase; FFAs: free fatty acids; FS%: percentage of fractional shortening; G6Pase: glucose-6-phosphatase; GRP78: glucose-regulated protein 78; HDL-c: high-density lipoprotein cholesterol; HFD: high-fat diet; HO-1: heme oxygenase-1; HOMA-IR: homeostatic model assessment for insulin resistance; ICAM-1: intercellular adhesion molecule-1; IL-1β: interleukin 1 beta; IL-6: interleukin 6; IκBα: inhibitor of nuclear factor kappa B; JNK: c-Jun N-terminal kinase; LCAT: lecithin-cholesterol acyltransferase; LDL-c: low-density lipoprotein cholesterol; LXRα: liver X receptor alpha; MMP-2: matrix metallopeptidase 2; MMP-9: matrix metallopeptidase 9; mRNA: messenger ribonucleic acid; MyHC: myosin heavy chain; NQO-1: NADPH quinone dehydrogenase 1; NRF2: nuclear factor erythroid 2–related factor 2; p62: sequestosome 1; p65: nuclear factor kappa B subunit; p-AKT: phosphorylated protein kinase B; p-AMPK: phosphorylated adenosine monophosphate-activated protein kinase; PARP: poly (ADP-ribose) polymerase 1; p-CREB: phosphorylated cAMP response element-binding protein; p-eIF2α: eukaryotic initiation factor 2 alpha; PEPCK: phosphoenolpyruvate carboxykinase; p-IR: phosphorylated insulin receptor; PLs: phospholipids; PPARα: peroxisome proliferator-activated receptor alpha; PPARγ: peroxisome proliferator-activated receptor gamma; PTP1B: protein tyrosine phosphatase 1B; SRB1: scavenger receptor class B type 1; SREBP1: sterol regulatory element binding transcription factor 1; SREBP2: sterol regulatory element binding transcription factor 2; TC: total cholesterol; TGF-β: transforming growth factor beta; TGs: triglycerides; TNF-α: tumor necrosis factor alpha; TUNEL: terminal deoxynucleotidyl transferase (TdT) dUTP nick-end labeling; VCAM-1: vascular cell adhesion molecule-1; ΔPUFA: thermally oxidized sunflower oil.

## 5. Summary and Conclusions

In the present review, we systematically compiled preclinical findings from in vitro and in vivo studies that explored the efficacy of curcumin derivatives to attenuate obesity and its associated metabolic complications. Furthermore, we also report the efficacy of these curcumin derivatives compared to curcumin. We identified eight synthetic curcumin derivatives (CNB-001, CDPP, CUR5-8, DHZ, C66, Salicylcurcumin, Y20, and Curc-mPEG454) that were shown to ameliorate obesity and metabolic dysregulation in animal models (Figure 4). Five of these (CNB-001, CDPP, CUR5-8, DHZ, and C66) were also reported to exhibit efficacy on obesity complications in cell culture models (Figure 4). In addition, five of the eight curcumin derivatives (CDPP, CUR5-8, Salicylcurcumin, DHZ, and Y20) reported here were shown to exhibit improved efficacy when compared to curcumin, even at lower doses (CDPP and Y20), while the other three curcumin derivatives were not effective compared to curcumin (CNB-001, C66, and Curc-mPEG454) (Figure 4). Our search of the literature demonstrated that these derivatives ameliorate obesity-induced metabolic diseases by modulating adipogenesis, lipid metabolism, insulin resistance, hepatic steatosis, lipotoxicity, inflammation, oxidative stress, endoplasmic reticulum stress, apoptosis, autophagy, fibrosis, and dyslipidemia to a greater extent than curcumin (Figure 4). These derivatives facilitate their efficacy using various mechanisms of action such as insulin signaling, AMPK and mTOR signaling, NF-κB/JNK and p38 MAPK signaling, and target various transcription factors including C/EBPα, CREB, PGC1α, PPARα, PPARγ, and SREBP1. Moreover, these curcumin derivatives mediate their effects by targeting circulating markers and tissues involved in obesity and metabolic dysfunction including adipose tissue, liver, kidney, skeletal muscle, and the heart (Figure 4).

Taken together, these studies demonstrate the efficacy of curcumin derivatives to ameliorate obesity and associated metabolic disorders. However, additional preclinical studies that compare the efficacy of these compounds to conventional anti-obesity drugs are required to substantiate the potential of these derivative compounds to serve as alternative therapeutics for obesity and related metabolic disorders. Furthermore, studies to elucidate the pharmacokinetic and safety profiles prior to clinical development as lead therapeutics are needed. It is worthy to note that curcumin derivatives such as CDPP and DHZ have been experimentally shown to exhibit better pharmacokinetic profiles than curcumin [78,99]. Similarly, various studies have been conducted to evaluate the safety of these curcumin derivatives. Like curcumin, CNB-001, for example, has been shown to exert neuroprotective effects by penetrating the blood–brain barrier while maintaining therapeutic safety [76,100]. Further studies are also required to establish standardized formulations of these lead synthetic curcumin derivatives and evaluate their safety and efficacy in clinical studies.

In conclusion, this review demonstrates the efficacy of synthetic curcumin derivatives to ameliorate obesity and obesity-related metabolic complications even more effectively compared to curcumin. These studies highlight the significance of synthetic curcumin derivatives as potential therapeutic agents to modulate obesity and associated metabolic dysfunction. Thus, we recommend more research studies to assess the bioavailability of curcumin derivatives using pharmacokinetic and pharmacodynamic approaches. In addition, randomized clinical trials are required to assess the clinical efficacy and safety of these derivatives to ameliorate obesity and its associated metabolic complications.

## Figures and Tables

**Figure 1 ijms-24-14366-f001:**
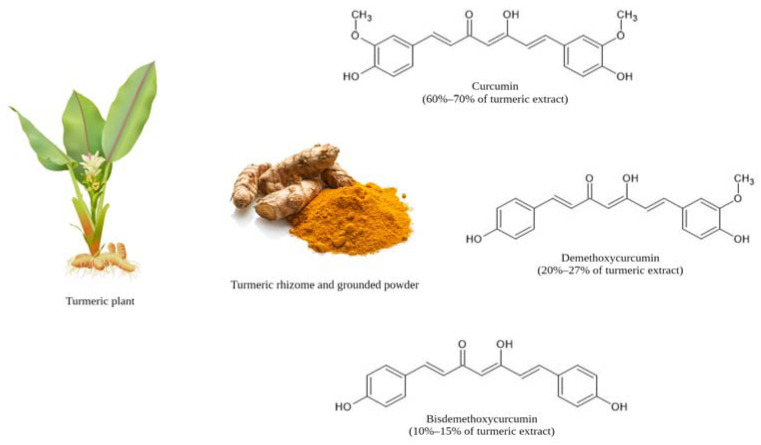
The turmeric (*Curcuma longa*) plant and rhizome, and the chemical structures of the curcuminoids including curcumin, demethoxycurcumin, and bisdemethoxycurcumin. Figure adapted and modified from Nelson et al. [39].

**Figure 4 ijms-24-14366-f004:**
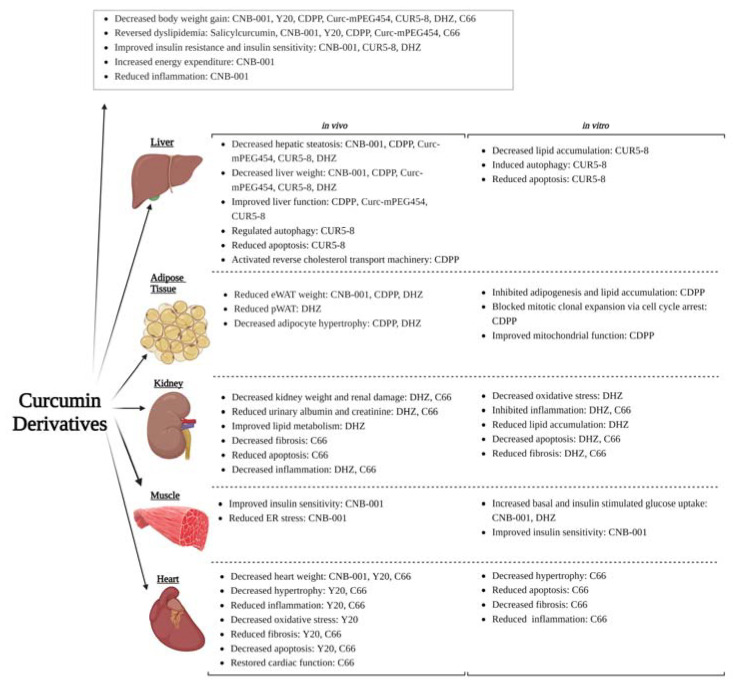
Schematic summary of the potential therapeutic effects of curcumin derivatives on obesity and its associated metabolic complications. The in vitro and in vivo studies summarized in the current review show that the curcumin derivatives exhibit their anti-obesity effects by modulating adipogenesis, lipid metabolism, insulin resistance, hepatic steatosis, lipotoxicity, inflammation, oxidative stress, endoplasmic reticulum stress, apoptosis, autophagy, fibrosis, and dyslipidemia. These effects are mediated by targeting circulating markers and tissues involved in obesity and metabolic dysfunction including the liver, white adipose tissue, kidney, skeletal muscle, and the heart. (Image created with BioRender.com, https://app.biorender.com/illustrations/6450d6bc57da0cda47ce8e88 accessed on 20 August 2023).

## Data Availability

No new data were created or analyzed in this study. Data sharing is not applicable to this article.

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
