# Peer review of "Therapeutic Effects of Curcumin Derivatives against Obesity and Associated Metabolic Complications: A Review of In Vitro and In Vivo Studies"

_ijms, 2023, doi:10.3390/ijms241814366_

Round 1
Reviewer 1 Report
Obesity and diabetes are associated with heart disease and are threatened by risk factors that shorten life expectancy. Therefore, the subject is significant.
1. It would have been useful to detail how the studies were selected for this literature review like the generalization and exclusion methods used in meta-analysis.
2. Review of literature that the use of anti -obesity drugs are infected with negative effects such as cardiovascular effects, stroke, cancer, and psychiatric disorders indeed indicate the need to detect materials from natural sources that have no similar effects. The effectiveness of curcumin derivatives to reduce obesity and its associated metabolic complications. Do not answer the question: If curcumin but not his metabolites can cross the blood barrier in the brain - whether the synthetic derivatives developed to cross this barrier cannot harm. What will we do in research and what is required in continuing research to ensure this?
Reviewer 2 Report
This manuscript is an in-depth review of the potential therapeutic effects of curcumin derivatives against obesity. The review is comprehensive, well-written and overall enjoyable to read. The different paragraphs, explicitly referencing the structural and pharmacokinetics ones, are correctly interleaved without excessive redundancy. Splitting some references into in vitro and in vivo is definitively an intelligent move to improve readability.
Some information on the identifiability of the reported research would have been of interest. More specifically, the authors described the effect as ‘improvement of insulin sensitivity’. However, the improvements found, while significantly different from control, are sometimes less than 5% of what could be obtained by more conventional drugs. Therefore, a personal assessment of the intensity of the putative overall metabolic improvement would have been a welcome addition.
The pharmacokinetic paragraph is essential, but the information is missing, such as the mathematical modelling after oral administration and values of micro constant extracted from the model, since these could be of interest for establishing a therapeutic scheme in patients.
